# Phenotype Analysis of Retinal Dystrophies in Light of the Underlying Genetic Defects: Application to Cone and Cone-Rod Dystrophies

**DOI:** 10.3390/ijms20194854

**Published:** 2019-09-30

**Authors:** Elise Boulanger-Scemama, Saddek Mohand-Saïd, Said El Shamieh, Vanessa Démontant, Christel Condroyer, Aline Antonio, Christelle Michiels, Fiona Boyard, Jean-Paul Saraiva, Mélanie Letexier, José-Alain Sahel, Christina Zeitz, Isabelle Audo

**Affiliations:** 1Institut de la Vision, Sorbonne Universités, UPMC Univ Paris 06, INSERM, CNRS, 17 rue Moreau, 75012 Paris, France; eboulanger@for.paris (E.B.-S.); saddekms@gmail.com (S.M.-S.); said.shamieh@gmail.com (S.E.S.); vanessa.demontant@inserm.fr (V.D.); christel.condroyer@inserm.fr (C.C.); aline.antonio@inserm.fr (A.A.); christelle.michiels@inserm.fr (C.M.); fiona.boyard@gmail.com (F.B.); j.sahel@gmail.com (J.-A.S.); christina.zeitz@inserm.fr (C.Z.); 2Fondation Ophtalmologique Adolphe de Rothschild, 75012 Paris, France; 3CHNO des Quinze-Vingts, DHU Sight Restore, INSERM-DHOS CIC1423, 28 rue de Charenton, 75012 Paris, France; 4Department of Medical Laboratory Sciences, Faculty of Health Sciences, Beirut Arab University, Beirut, Lebanon; 5IntegraGen SA, Genopole CAMPUS 1 bat G8 FR, 91030 Evry, France; jean-paul.saraiva@integragen.com (J.-P.S.); melanie.letexier@integragen.com (M.L.); 6Académie des Sciences-Institut de France, 75006 Paris, France; 7Department of Ophthalmology, The University of Pittsburgh School of Medicine, Pittsburg, PA 15213, USA; 8University College London Institute of Ophthalmology, 11-43 Bath Street, London EC1V 9EL, UK

**Keywords:** cone-rod dystrophy, next-generation sequencing

## Abstract

Phenotypes observed in a large cohort of patients with cone and cone-rod dystrophies (COD/CORDs) are described based on multimodal retinal imaging features in order to help in analyzing massive next-generation sequencing data. Structural abnormalities of 58 subjects with molecular diagnosis of COD/CORDs were analyzed through specific retinal imaging including spectral-domain optical coherence tomography (SD-OCT) and fundus autofluorescence (BAF/IRAF). Findings were analyzed with the underlying genetic defects. A ring of increased autofluorescence was mainly observed in patients with *CRX* and *GUCY2D* mutations (33% and 22% of cases respectively). “Speckled” autofluorescence was observed with mutations in three different genes (*ABCA4* 64%; *C2Orf71* and *PRPH2*, 18% each). Peripapillary sparing was only found in association with mutations in *ABCA4*, although only present in 40% of such genotypes. Regarding SD-OCT, specific outer retinal abnormalities were more commonly observed in particular genotypes: focal retrofoveal interruption and *GUCY2D* mutations (50%), foveal sparing and *CRX* mutations (50%), and outer retinal atrophy associated with hyperreflective dots and *ABCA4* mutations (69%). This study outlines the phenotypic heterogeneity of COD/CORDs hampering statistical correlations. A larger study correlating retinal imaging with genetic results is necessary to identify specific clinical features that may help in selecting pathogenic variants generated by high-throughput sequencing.

## 1. Introduction

Cone and cone-rod dystrophies (COD/CORDs) refer to a heterogeneous group of inherited retinal disorders (IRDs), characterized predominantly by cone impairment. They are the most common cause of hereditary cone dysfunction, with a prevalence of 1:40,000 [1]. They are characterized by progressive central vision loss, photophobia, and color vision abnormalities in childhood or early adulthood. In most cases, with a variable onset in the course of the disease, patients develop secondary rod system involvement that leads to night blindness and peripheral visual field constriction [1]. On fundoscopy, the macular appearance ranges from normal to bull’s eye maculopathy, or more severe macular atrophy with possible pigmentary changes in the periphery in the case of rod photoreceptor involvement [2]. Full-field electroretinogram (ERG) examination is key in diagnosis and reveals both cone and rod impairment with predominant cone dysfunction. In advanced cases, ERG responses may be undetectable, making the distinction between CORD and severe rod-cone dystrophy (RCD) difficult, and somewhat artificial, in these cases. Progressive COD/CORDs need to be distinguished from cone dysfunction syndromes, which are stationary and congenital with normal rod function [3]. COD/CORDs often present as an isolated disease, but they can also be part of a syndrome as in Bardet–Biedl or Jalili syndromes or spinocerebellar ataxia 7 [1,4,5].

The genetic basis of COD/CORDs is highly heterogeneous, with significant overlap with other IRD-associated gene defects. All modes of inheritance have been reported: autosomal recessive (ar), autosomal dominant (ad), or X-linked (xl). To date, mutations in 33 genes have been implicated in COD/CORDs (https://sph.uth.edu/retnet/ September 2017). Mutations in *ABCA4* (ATP-binding cassette, sub-family A, member 4) [6,7], *GUCY2D* (guanylate cyclase 2D) [8,9], and *RPGR* (retinitis pigmentosa GTPase regulator) [10,11] are major causes of ar, ad, and xl COD/CORDs, respectively.

Because of this high phenotypic and genetic heterogeneity, patient management and genetic counseling are currently challenging for clinicians who diagnose CODs and CORDs. The application of high-throughput sequencing tools for genetic diagnosis increases the yield to identify the underlying genetic defect(s), but filtering and interpretation of massive data are necessary to identify causative gene defects. The purpose of this cross-sectional study was to conduct a phenotyping analysis of a large cohort of COD/CORDS patients, previously genetically investigated by a targeted next-generation sequencing (NGS) panel, [12] using retinal imaging; therefore, we attempted to identify schematic phenotype/genotype associations in order to help in interpreting massive data generated from NGS.

## 2. Results

### 2.1. Clinical and Genetic Characteristics

Baseline clinical characteristics of all probands are summarized in Table 1. 

Sporadic, ar, and ad (43%, 38%, and 19%, respectively) cases were included in the study. As previously reported [12], applying a targeted NGS approach, pathogenic or likely pathogenic mutations were identified in 23 genes (12 known COD/CORDs-associated genes and 11 other retinal disease associated genes) (Table 2, detailed genetic results and co-segregation analysis are available in a previous article [12]). Mutations in *ABCA4* (ATP-binding cassette, sub-family A, member 4) and *GUCY2D* (guanylate cyclase 2D) were reported in 36% and 30% of the individuals showing ar and ad modes of inheritance, respectively, making them the major defective genes in our cohort. 

### 2.2. Retinophotography

The fundus revealed abnormalities either limited to the macular area (64.5%) or extended to the peripheral retina (37.5%). Four distinct fundus patterns were identified: macular RPE alterations (21%), “bull’s eye maculopathy” with perifoveal atrophy sparing the fovea (9%), macular atrophy (34%), and extensive retinal atrophy (36%). (Figure 1) 

### 2.3. SD-OCT 

Table 3 shows abnormalities observed in SD-OCT, illustrated by Figure 2.

ELM, EZ, and IZ were disrupted in the foveal/perifoveal region in 80%, 94%, and 100% of cases, respectively. Foveal sparing of the outer retinal layers was observed in 6 patients (12%), whereas hyporeflective foveal cavitation was observed in 3 patients (6%). Hyper-reflective deposits above the RPE were observed in 28% of cases. Macular outer nuclear layer atrophy was present in 73% of patients. Among these, 41% revealed extensive atrophy beyond the vascular arcades.

In 2 cases, SD-OCT revealed hyporeflective macular cysts in the outer and inner nuclear layers, without macular edema.

### 2.4. FAF

Table 4 shows abnormal patterns observed in BAF and IRAF, illustrated in Figure 3.

BAF revealed abnormalities either limited to the macular area (31%) or extended to the peripheral retina (68%), and 6 distinct patterns were identified: discrete foveal abnormalities; macular hypo-autofluorescent (hypoAF) area surrounded, or not, by a ring of hyper-autofluorescence (hyperAF) or macular hypoAF spots sparing the peripheral retina; macular hypoAF area associated with hypoAF spots or “speckled” pattern in the peripheral retina; and extensive and confluent hypoAF patches associated, or not, with a “speckled” pattern.

A macular hyperAF ring was observed in 23% of patients, which included the optic nerve in 7% of cases. Among the 38 patients with extensive BAF abnormalities, 16% (6/38) revealed a peripapillary sparing.

IRAF revealed abnormalities either limited to the macular area (44%) or extended to the peripheral retina (56%), and 5 distinct patterns were identified: discrete foveal abnormalities, macular hypoAF area or macular hypoAF spots sparing the peripheral retina, macular hypoAF area associated with hypoAF spots in the peripheral retina, and extensive macular and peripheral hypoAF.

Among the 29 patients with extensive IRAF abnormalities, 6 (21%) revealed a peripapillary sparing, which was also observed in BAF. The macular hyperAF ring was detected on both BAF and IRAF (excepted for 1 patient whose IRAF imaging was not performed).

Unlike BAF imaging, the “speckled” pattern was not observed with IRAF in this cohort, and this pattern in BAF corresponded to confluent hypoAF areas in IRAF. 

### 2.5. Genotype Analysis with Respect to Structural Abnormalities

Figure 4 shows the profile of mutated genes for each feature observed on retinophotography, SD-OCT, and FAF.

On SD-OCT, focal retrofoveal interruption of the outer retinal layers was more commonly observed in patients with *GUCY2D* mutations (50%), whereas foveal sparing of the outer retinal layers was mainly associated with *CRX* mutations (50%). Among patients with hyper-reflective deposits above the RPE, 69% had *ABCA4*.

Regarding FAF, a ring of increased AF was mainly observed in patients with *CRX* and *GUCY2D* mutations (33% and 22% of cases, respectively). The “speckled” AF was present in patients with mutations in 3 different genes (*ABCA4* 64%; *C2Orf71* and *PRPH2*, 18% each). In addition, a peripapillary sparing was only found in patients with *ABCA4* mutations. (Figure 5).

## 3. Discussion

We investigated a large series of CODs and CORDs patients, and we carefully evaluated their clinical characteristics using retinal imaging, which were further related to the genetic diagnosis.

### 3.1. Retinophotography

Fundus abnormalities ranged from discrete macular RPE alterations to extensive chorioretinal atrophy. (Figure 1) “Bull’s eye” maculopathy, defined by perifoveal atrophy sparing the fovea, was observed only in 9% of cases, whereas macular atrophy was present in 34% of cases. These observations are similar to those reported by Thiadens et al. in a longitudinal study of 239 patients [2]. After 10 years of follow-up, 35% of CODs and 58% of CORDs showed macular atrophy, including “bull’s eye” maculopathy.

In this cohort, fundus alterations limited to the macula were mainly observed in patients with *ABCA4, GUCY2D, PROM1, CRX, SEMA4A,* and *PDE6C* mutations. But these same mutated genes (except for *GUCY2D*) were also implicated in more severe clinical presentations with extensive retinal atrophy. These results outline the phenotypic variability associated with mutations in the same gene or even a same mutation. In a study including five families with autosomal dominant COD/CORDS cases carrying the same heterozygous p. (R373C) *PROM1* exchange, phenotypes ranged from isolated macular dysfunction with “bull’s eye” maculopathy to severe generalized cone-rod or rod-cone dysfunction, in patients from the same family [13]. In another study including 18 patients from 11 families, multimodal retinal imaging combined with electrophysiology identified 4 Leber congenital amaurosis (LCA), 2 retinitis pigmentosa (RP), 6 COD/CORDs, and 6 “atypical maculopathy” patients carrying the same heterozygous *CRX* mutation [14].

In the present study, unlike mutations in *ABCA4, PROM1,* or *CRX*, *GUCY2D* mutations were always associated with fundus alterations solely restricted to the macula (i.e., subtle changes or round atrophic lesion). No report of diffuse fundus alterations associated with *GUCY2D* mutations was found in the literature [15]. Thus, in this cohort, a *GUCY2D* mutation could be suspected in the case of restricted alterations to the macula on fundoscopy, but it is unlikely in case of peripheral damage.

### 3.2. SD-OCT

Outer hyper-reflective band abnormalities were more frequent in the foveal/perifoveal regions, where cone density is the highest. At a more advanced stage, the outer nuclear layer was atrophic with underlying choroidal hyper-reflectivity by window defect, limited to the macular area or extended to the peripheral retina. (Figure 2)

#### 3.2.1. Outer Hyperreflective Bands 

In a recent study, outer retinas of 12 CORDs patients with macular atrophy on fundoscopy were analyzed using SD-OCT [16]. Foveal IZ, EZ, and ELM were disrupted in 100%, 92%, and 83% of cases, respectively, whereas the RPE layer was always preserved. Outside the foveal region, IZ was absent in 100% of cases, whereas EZ was still present with decreased intensity. Our study also found a disruption of the outer hyperreflective bands, more frequently observed in the foveal and perifoveal regions, and was predominant on the IZ followed by EZ and ELM.

According to Lima et al., the lack of visibility of the IZ on SD-OCT could be linked to the loss or the shortening of the cone outer segments, reducing the possibility of proper interdigitation with the apical processes of the RPE cells [16]. Similarly, in COD animal models, there is a loss of the inner and outer cone segments, or there are alterations in the anatomic configuration of the apical processes [17,18]. This hypothesis is uncertain because there is a lack of correlative human histology. According to Inui et al., the loss of the IZ at the foveal region would be an early sign of cone-dominant photoreceptor impairment [19]. However, this feature is not specific to cone dystrophy, as it can be observed in patients with achromatopsia, high myopia, and age-related macular degeneration [20,21]. 

In this cohort, 50% of patients (N = 6) with selective retrofoveal abnormalities (Figure 2a) revealed *GUCY2D* mutations, whereas 50% of patients (N = 6) with foveal sparing (Figure 2g) carried *CRX* mutations.

#### 3.2.2. Hyporeflective Foveal Cavitation 

In three patients, a hyporeflective cavitation was observed in the foveal outer retina. (Figure 2b) This feature was previously described in COD/CORDs, congenital achromatopsia, and blue cone monochromacy [22]. It would be explained by the lack of cone outer segments with intact ELM [19,23]. According to Leng et al., foveal cavitation is a characteristic feature of cone dysfunction syndromes, without being specific to one disorder [24]. This feature seems to be associated with different genetic defects. In this cohort, it was observed in patients with *GUCY2D*, *EYS,* and *SEMA4A* mutations, whereas in the literature, it is described in patients with defects in *GUCY2D, GUCA1A,* and *ABCA4* [15,24].

#### 3.2.3. Hyper-reflective Deposits above the RPE 

In 28% of patients, small hyperreflective dots or deposits were observed above the RPE, predominantly in the foveal and perifoveal regions. (Figure 2e) These deposits could represent lipofuscin or accumulated debris from photoreceptor outer segments that have not been properly phagocytosed by impaired RPE cells and/or microglial cells. This hypothesis lacks correlative human histology. In almost 70% of cases, this pattern was observed in patients with *ABCA4* mutations. This could be explained by the role of the resultant protein as an active transporter, therefore leading to retinoid accumulation above and inside RPE cells when mutated [25].

#### 3.2.4. Hyporeflective Macular Cysts 

Two patients with *CRX* and *CRB1* mutations showed macular hyporeflective cysts on SD-OCT, at the level of the outer and inner nuclear layers. (Figure 2f) These cysts were not associated with macular edema and seemed to be degenerative within the atrophic retina. While hyporeflective cysts are commonly observed in RP, their association with COD/CORDs has not been reported so far [16,26].

### 3.3. Autofluorescence

#### 3.3.1. BAF

According to Wang et al., BAF imaging of COD/CORDs is highly heterogeneous and not specific to this pathology [27]. In this study, 6% of patients revealed an increase in foveal AF (loss of physiological foveal hypoAF or foveal hyperAF), which could reflect cone-RPE dysfunction. However, most of the patients showed macular or diffuse hypoAF, related to irreversible degeneration of photoreceptor and RPE cells. A recent study correlated the surface of BAF abnormalities using ultrawide field imaging with functional data (visual acuity, ERG, visual field). The authors showed that the extent of abnormal BAF was directly correlated to the severity of functional impairment [28].

In this cohort, patients with BAF abnormalities restricted to the macular region carried *GUCY2D*, *CRX*, *PDE6C*, and *SEMA4A* mutations. For some patients with *ABCA4* and *PROM1* mutations, abnormalities were more extensive in BAF than expected on fundoscopy. This observation outlines the fact that BAF imaging is more sensitive than retinophotography for the evaluation of structural impairment [28]. Gelman et al. recently showed that BAF features in Stargardt were able to discriminate Stargardt group 1 from group 2 as a functional correlate [29].

#### 3.3.2. HyperAF Ring 

In almost 25% of patients, macular hypoAF was surrounded by a ring of hyperAF. (Figure 3b) This feature is not specific to COD/CORDs, as it can be observed in other inherited retinal dystrophies, particularly in RP (59–94% of cases) [30,31]. It is reported to reflect the transition between abnormal and normal retina. However, in contrast to RP, in COD/CORDs, the outer retina outside the ring is well-preserved, whereas it is atrophic within the ring. The hyperAF may suggest lipofuscin accumulation secondary to an increased rate of photoreceptor outer segment phagocytosis, with a direct toxicity on photoreceptor and RPE cells [32]. It could also be due to a window defect on RPE autofluorescence secondary to the lack of outer retinal layers. A correlation between SD-OCT and BAF would document the later hypothesis.

In the cohort, the hyperAF ring was observed in patients with *CRX* or *GUCY2D* mutations in 50% of cases, which is consistent with literature [14,33]. Associations with *GUCA1A* mutations have also been reported, but in our cohort, the only patient with a mutation on this gene revealed an extensive retinal atrophy. This phenotype is much more severe than usually reported in the literature in the case of mutations in this gene [15].

Thirty percent of patients, carrying mutations in *SEMA4A*, *AIPLI*, *NMNAT1*, or *RDH12*, displayed a hyperAF ring surrounding the optic nerve. This feature is also reported with X-linked retinal dystrophies in association with mutations in *RPGR* [3]. These patients were not included in the present study, as explained earlier [12].

#### 3.3.3. “Speckled” Pattern 

A “speckled” or “mottled” pattern was observed in 20% of patients, either in the macular region or extending beyond the vascular arcades. (Figure 3e,g) This feature with alternating hyperAF or hypoAF spots could correspond with co-existing window defects on RPE AF, lipofuscin deposits, and RPE atrophy [34,35,36]. This “speckled” feature is known to be associated with *ABCA4* and *PRPH2* mutations [3,34]. In this study, it was also observed in patients with mutations of three different genes: *ABCA4* (64%), *PRPH2* (18%), and also *C2Orf71* (18%). However, detailed semiologic observation revealed different features for each gene. (Figure 6) Patients with *PRPH2* mutations led to larger, coarser, and less numerous hyperAF spots, with a reticular pattern, surrounding macula and the optic nerve (Figure 6a). For patients with *ABCA4* mutations, hypoAF and hyperAF spots were much more numerous and widespread, sparing classically the peripapillary region (Figure 6b). Finally, spots observed in patients with *C2Orf71* mutations were smaller, leading to a granular pattern, extending beyond the vascular arcades without peripapillary sparing (Figure 6c). Such associations need to be confirmed by statistical analysis in a larger-scale study.

#### 3.3.4. Peripapillary Sparing 

Sparing of the peripapillary AF was classically considered as pathognomonic and constant in *ABCA4*-related diseases, even at the late stages with extensive retinal atrophy [36]. Recent studies showed that, although being specific, peripapillary sparing is not constant [37,38,39,40]. (Figure 3g,h)

A retrospective study including 32 patients with Stargardt groups 1, 2, and 3, according to the functional classification, showed the absence of peripapillary sparing in 6.7%, 100%, and 90% of cases, respectively [37]. These results were probably overestimated because only one heterozygous *ABCA4* variant was necessary for the inclusion in the study, suggesting that *ABCA4* mutations may not have been the gene defect for some of the patients. In our study, peripapillary sparing was exclusively observed in patients carrying homozygous or compound heterozygous *ABCA4* mutations. However, it was observed in only 40% of *ABCA4-*mutated patients with diffuse BAF abnormalities. Cideciyan et al. hypothesized that in the peripapillary region, turn-over of photoreceptor outer segments is reduced, with an optimized photoreceptor/RPE cells ratio explaining AF sparing. Of note, in *MERTK*-related diseases, characterized by outer segment phagocytic defects, there are widespread severe retinal abnormalities with no peripapillary sparing [41]. Another hypothesis would be the protective effect of the thickened retinal nerve fibers in this particular region towards oxidative stress and lipofuscin accumulation secondary to light exposure [36].

### 3.4. IRAF

IRAF imaging documents the distribution of melanin and melanolipofusin granules within RPE cells, as well as melanin within choroidal melanocytes [42]. In our study, all patients showed hypoAF in IRAF, with various patterns, limited to the macular region (44%) or extended to the peripheral retina (56%). Reduced IRAF could be explained by several mechanisms: posterior displacement of melanin granules secondary to lipofuscin accumulation in RPE cells, alteration of the RPE cell phagocytic activity, or loss of RPE cells [42].

No patient from the cohort revealed any increased IRAF or “speckled” aspect in IRAF. HyperAF spots in BAF appeared as hypoAF in IRAF in all patients (Figure 3e). In a study including 16 patients with *ABCA4*-related retinal dystrophies, most of the patients (9/16) also revealed reduced IRAF, whereas for the other patients (4/16), a few small hyperAF dots in IRAF corresponding to hyperAF dots in BAF were observed [34]. Increased IRAF could be explained by apical displacement of melanin granules secondary to lipofuscin accumulation in RPE cells or melanin/melanolipofuscin accumulation secondary to increased RPE cell phagocytic activity [42].

Peripapillary preservation of IRAF was present in all six patients with peripapillary preservation of BAF (Figure 3h). Kellner et al. also found peripapillary sparing in 4/16 patients in both BAF and IRAF imaging [34]. These observations suggest a good correlation between BAF and IRAF imaging for the detection of peripapillary sparing.

## 4. Patients and Methods

### 4.1. Study Population

In a previous study [12], targeted NGS using a panel of 123 genes implicated in inherited retinal diseases (IRDs) was applied to 95 genetically unrelated ad and ar cone and cone-rod dystrophy cases, based on functional abnormalities, leading to the identification of the underlying genetic defects in 62.1% (59/95) of cases. Among the resolved cases, 58 subjects were selected for structural abnormalities analysis. 

### 4.2. Clinical Examination

Each patient underwent full ophthalmic examination, as described earlier [43]. Clinical assessment was completed by retinophotography (CR1, Canon, Tokyo, Japan), spectral-domain optical coherence tomography (SD-OCT), and short wavelength (i.e., blue BAF) and near-infrared (IRAF) fundus autofluorescence imaging (FAF) (Spectralis^®^ OCT and HRAII^®^, Heidelberg Engineering, Dossenheim, Germany, respectively). Outer hyper-reflective bands were analyzed, including the external limiting membrane (ELM), the ellipsoid zone (EZ), the interdigitation zone (IZ), and the retinal pigment epithelium (RPE) [44,45]. The inner retina was further analyzed.

The study protocol adhered to the tenets of the Declaration of Helsinki and was approved by the local ethics committee (CPP, Comité de Protection des Personnes Ile de France V).

## 5. Conclusions

This study outlines the high phenotypic and genetic heterogeneity of these rare diseases with a small sample size for each genotype. For this reason, we were not able to perform statistical association/correlation analysis. However, we attempted to draw schematic orientations only applicable to our cohort, based on retinal imaging abnormalities and genetic results. A larger study with a control group is necessary to extract statistically clinical patterns that may help in selecting pathogenic variants generated by high-throughput sequencing in our clinical practice. Second, we performed a cross-sectional study and did not correlate our findings to the disease stage. Longitudinal follow-up on a larger group of patients will help refine the identified correlations.

In conclusion, there is a need for the best genetic and phenotypic characterization of IRDs, applying, for instance, multimodal retinal imaging to identify specific phenotype–genotype correlations in order to improve diagnosis, management, genetic counseling, and patient information regarding the prognosis of IRDs. Furthermore, with the recent development of new therapeutic approaches, these results will also allow the clinician to guide therapeutic choices and establish clinical outcomes to monitor treatment efficacy.

## Figures and Tables

**Figure 1 ijms-20-04854-f001:**
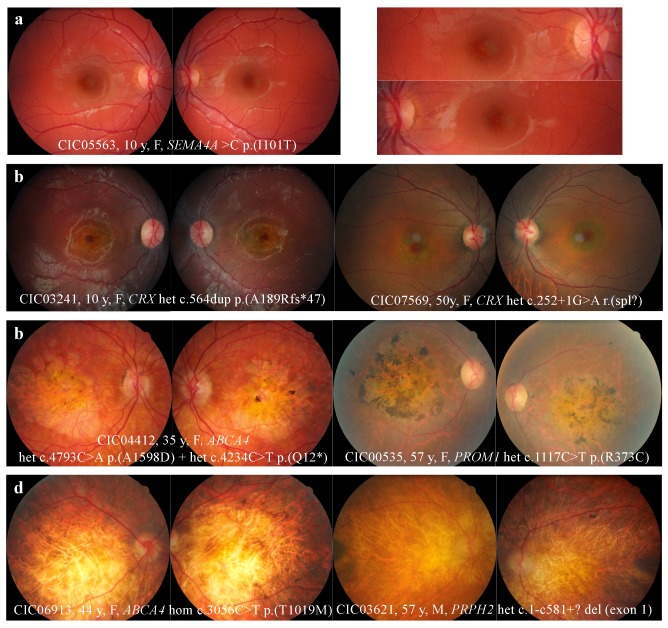
Fundus abnormalities observed in cone and cone-rod dystrophy (COD/CORD) patients of the cohort. (**a**) Macular retinal pigment epithelium (RPE) alterations. (**b**) “Bull’s eye maculopathy” defined by perifoveal atrophy sparing the fovea. Note the temporal pallor of the optic disc for patient CIC03241, a clinical feature known to be associated with CODs. (**c**) Retinal and RPE atrophy limited to the macular region. Note the pigmented aspect above the macular atrophy, sharply marked in patient CIC00535. (**d**) Extensive retinal atrophy from the macula to the peripheral retina. Note the optic disk pallor, narrowing vascular network, and peripheral osteoblasts evoking the differential diagnosis of retinitis pigmentosa.

**Figure 2 ijms-20-04854-f002:**
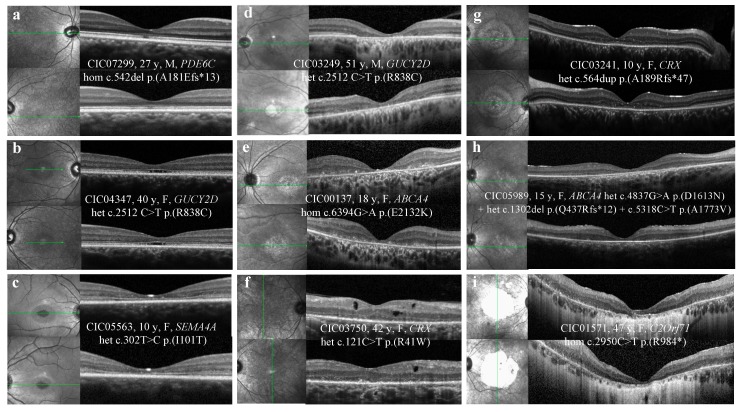
SD-OCT abnormalities observed in COD/CORDs patients of the cohort. (**a**) Abnormalities limited to the foveal region, irregular aspect or disruption of the ellipsoid zone (EZ) and the interdigitation zone (IZ). (**b**) Hyporeflective foveal cavitation. EZ and IZ are disrupted, while ELM and RPE layers are respected. (**c**) Perifoveal and foveal abnormalities. EZ and IZ are disrupted, while ELM is respected. (**d**) Outer retinal atrophy of the foveal and perifoveal regions. (**e**) Hyper-reflective deposits above the RPE in the foveal and perifoveal regions. (**f**) Hyporeflective cysts at the level of the outer and inner nuclear layers without macular edema. (**g**) Foveal sparing of the outer hyper-reflective layers; visual acuity is quite preserved for this patient (20/63 OD, 20/80 OG). (**h**) Outer retinal atrophy of the macular region. (**i**) Extensive chorioretinal atrophy with retinal thinning of the foveal region and choroidal hyperreflectivity by window defect. SD-OCT: spectral-domain optical coherence tomography; COD/CORDs: cone and cone-rod dystrophy; EZ: ellipsoid zone; IZ: interdigitation zone; ELM: external limiting membrane; RPE: retinal pigment epithelium.

**Figure 3 ijms-20-04854-f003:**
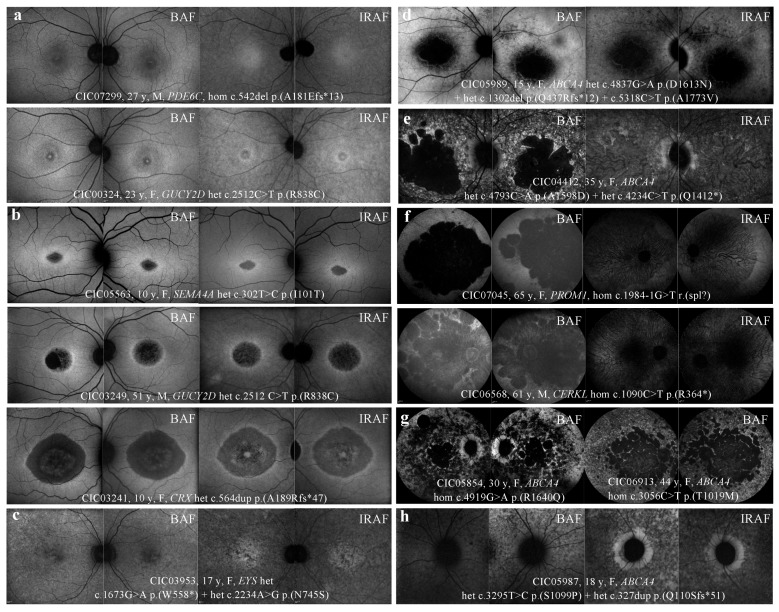
FAF and IRAF patterns observed in COD/CORD patients of the cohort. (**a**) Foveal and perifoveal hyperAF in BAF; foveal and perifoveal hypoAF in IRAF. (**b**) Macular hypoAF surrounded by a hyperAF ring in BAF and IRAF. (**c**) Macular hypoAF spots in BAF and IRAF. (**d**) Macular hypoAF associated with peripheral hypoAF spots in BAF and IRAF. (**e**) Macular hypoAF associated with peripheral “speckled” aspect (alternating hypoAF and hyperAF spots) in BAF and confluent hypoAF spots in IRAF. (**f**) Confluent hypoAF areas, involving the optic disc and extending beyond the vascular arcades. (**g**) Extensive and confluent hypoAF areas with a “speckled” aspect, associated (**left**) or not (**right**) with peripapillary sparing. (**h**) Sparing of peripapillary autofluorescence, more pronounced in IRAF than in BAF in this case. HypoAF: hypo-autofluorescence/hypo-autofluorescent; HyperAF: hyper-autofluorescence/hyper-autofluorescent; BAF: blue autofluorescence; IRAF: infrared autofluorescence.

**Figure 4 ijms-20-04854-f004:**
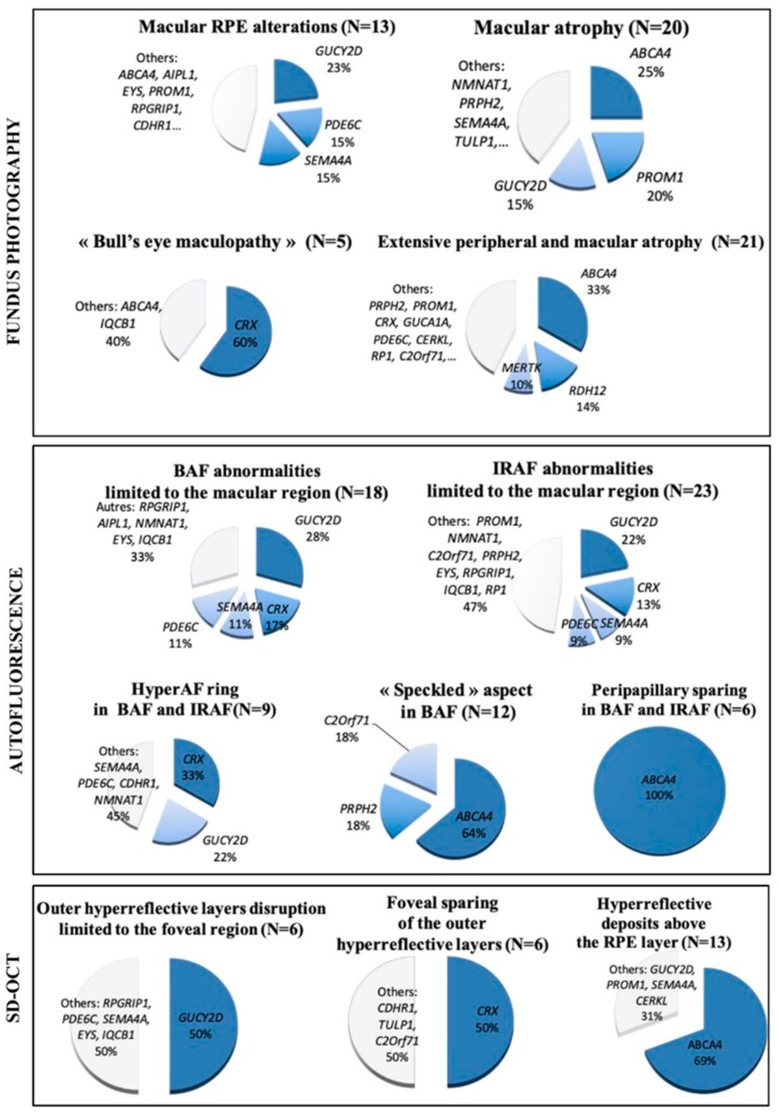
Profile of mutated genes for each feature observed on retinophotography, fundus autofluorescence, and spectral domain OCT.

**Figure 5 ijms-20-04854-f005:**
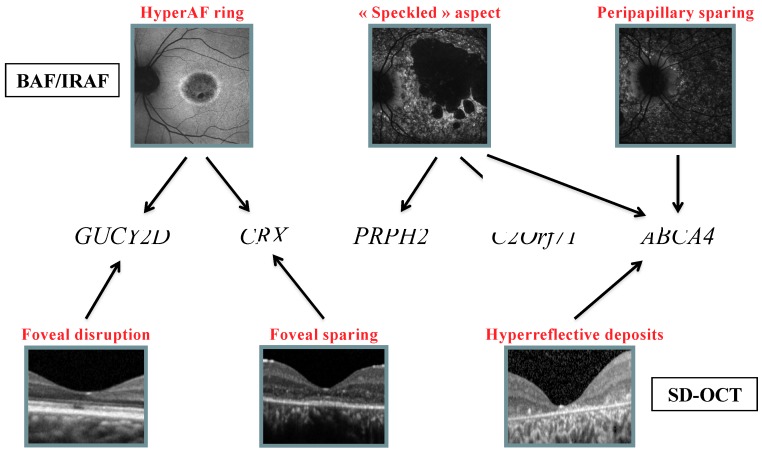
Decision tree showing clinical features of FAF and SD-OCT that may help to identify the causative genetic defect.

**Figure 6 ijms-20-04854-f006:**
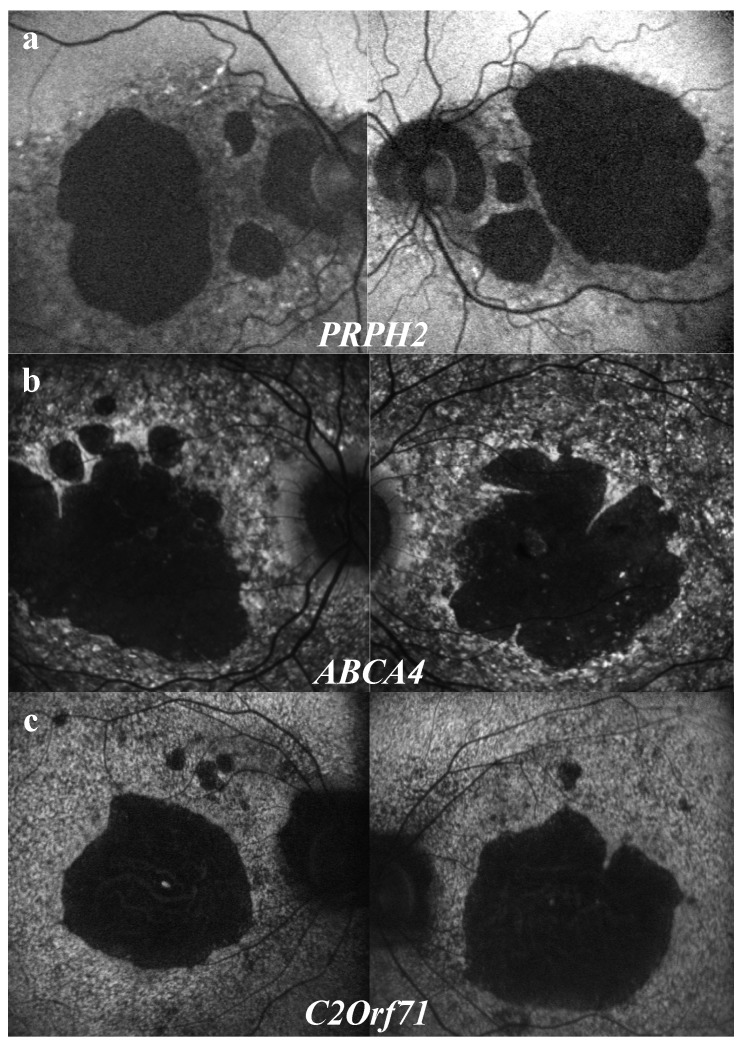
Characteristic features of the “speckled” aspect depending on the gene defect. (**a**) *PRPH2* mutations: hyperAF spots are larger and less numerous, with reticular aspect, surrounding macular atrophy and the optic nerve. (**b**) *ABCA4* mutations: hyperAF spots are more numerous and widespread, sparing classically the peripapillary region. (**c**) *C2Orf71* mutations: hyperAF are smaller, leading to a granular aspect, extending beyond the vascular arcades without peripapillary sparing.

**Table 1 ijms-20-04854-t001:** Clinical characteristics of the cohort (N = 58 patients).

Age at the onset of symptoms (decreased central vision and/or photophobia) (median (min–max), years)	10 (1–55)	(N = 36) *
Age at clinical exam (average (SD, min–max), years)	35 (5, 7–78)	
Mode of inheritance		(N = 58)
Autosomal recessive, n (%)	22 (38)	
Autosomal dominant, n (%)	11 (19)	
Sporadic, n (%)	25 (43)	
Sex		(N = 58)
Male, n (%)	14 (24)	
BCVA (average (SD), ETDRS)	20:400 (20:125)	(N = 53)
BCVA (average (SD), LogMar)	1.3 (0.8)	
Spherical equivalent (average (SD), Diopters)	−2.5 (3.5)	(N = 49)

DS: standard deviation; BCVA: best corrected visual acuity; ETDRS: Early Treatment Diabetic Retinopathy Study. * Data were not collected for the 22 other patients.

**Table 2 ijms-20-04854-t002:** Summary of 58 patients carrying pathogenic and likely pathogenic mutations in known CCRD genes or other retinal disease genes.

ID	Type	Gene	Allele Status		cDNA	Protein	References
***Known CCRD genes***
**CIC00137**	sporadic	*ABCA4*	Ho	47	c.6394G>A	p.(E2132K)	(Boulanger-Scemama et al. 2015)
**CIC00162**	Ar	*ABCA4*	Het	31	c.4546_4547del	p.(Q1516Afs*38)	(Boulanger-Scemama et al. 2015)
		*ABCA4*	Het	16	c.2463G>A	p.(W821*)	(Boulanger-Scemama et al. 2015)
**CIC00765**	Ar	*ABCA4*	Ho	47	c.6445C>T	p.(R2149*)	(Lewis et al. 1999)
(rs61750654)
**CIC03436**	Ar	*ABCA4*	Ho	42	c.5892del	p.(G1965Efs*9)	[1]
**CIC04412**	sporadic	*ABC4A*	Het	34	c.4793C>A	p.(A1598D)	(Maugeri et al. 2000)
(rs61750155)
		*ABCA4*	Het	28	c.4234C>T	p.(Q1412*)	(Maugeri et al. 2000)
(rs61750137)
**CIC04645**	Ar	*ABCA4*	Ho	13	c.1924T>C	p.(F642L)	(Boulanger-Scemama et al. 2015), but c.1924T>A p.F642I in (Jin et al. 2014)
**CIC05087**	sporadic	*ABCA4*	Ho	IVS 11	c.1554+1G>C	r.(spl?)	(Boulanger-Scemama et al. 2015)
**CIC05853**	sporadic	*ABC4A*	Ho	22	c.3259G>A	p.(E1087K)	(Allikmets et al. 1997)
(rs61751398)
**CIC05854**	Ar	*ABC4A*	Ho	35	c.4919G>A	p.(R1640Q)	(Simonelli et al. 2000)
(rs61751403)
**CIC05987**	Ar	*ABC4A*	Het	22	c.3295T>C	p.(S1099P)	(Fumagalli et al. 2001)
(rs61750119)
		*ABC4A*	Het	4	c.327dup	p.(Q110Sfs*51)	(Boulanger-Scemama et al. 2015)
(rs 61748531)
**CIC05989**	sporadic	*ABC4A*	Het	34	c.4837G>A	p.(D1613N)	(Boulanger-Scemama et al. 2015)
		*ABC4A*	Het	10	c.1302del	p.(Q437Rfs*12)	(Boulanger-Scemama et al. 2015)
		*ABCA4*	Het	38	c.5318C>T	p.(A1773V)	(Stenirri et al. 2008)
**CIC06170**	sporadic	*ABC4A*	Het	44	c.6089G>A	p.(R2030Q)	(Lewis et al. 1999)
(rs61750641)
		*ABC4A*	Het	IVS 24	c.3607+3A>T	r.(spl?)	(Boulanger-Scemama et al.2015)
		*ABCA4*	Het	14	c.2034G>T	p.(K678N)	(Huang et al. 2014)
**CIC06694**	sporadic	*ABC4A*	Het	IVS36	c.5196+1G>A	r.(spl?)	(Kitiratschky et al. 2008)
		*ABC4A*	Het	22	c.3322C>T	p.(R1108C)	(Briggs et al. 2001)
**CIC06735**	Ar	*ABC4A*	Ho	42	c.5892del	p.(G1965Efs*9)	[1]
**CIC06913**	Ar	*ABCA4*	Ho	21	c.3056C>T	p.(T1019M)	(Rozet et al. 1998)
(rs201855602)
**CIC04239**	Ar	*CDHR1*	Ho	9	c.838C>T	p.(R280*)	(Boulanger-Scemama et al. 2015)
**CIC06568**	Ar	*CERKL*	Ho	8	c.1090C>T	p.(R364*)	Thesis (Sergouniotis P. 2012) [2]
**CIC07299**	sporadic	*PDE6C*	Ho	2	c.542del	p.(A181Efs*13)	(Boulanger-Scemama et al. 2015)
**CIC05563**	Ad	*SEMA4A*	Het	4	c.302T>C	p.(I101T)	(Boulanger-Scemama et al. 2015)
(rs149652495)
**CIC07563**	sporadic	*SEMA4A*	Ho	3	c.241C>T	p.(R81*)	(Boulanger-Scemama et al. 2015)
**CIC00324**	Ad	*GUCY2D*	Het	13	c.2512C>T	p.(R838C)	(Kelsell et al. 1998)
(rs61750172)
**CIC03249**	Ad	*GUCY2D*	Het	13	c.2512C>T	p.(R838C)	(Kelsell et al. 1998)
(rs61750172)
**CIC04347**	Ad	*GUCY2D*	Het	13	c.2512C>T	p.(R838C)	(Kelsell et al. 1998)
(rs61750172)
**CIC04918**	Ad	*GUCY2D*	Het	13	c.2512C>T	p.(R838C)	(Kelsell et al. 1998)
(rs61750172)
**CIC00597**	sporadic	*GUCY2D*	Het	14	c.2747T>C	p.(I916T)	(De Castro-Miró et al. 2014)
**CIC06352**	sporadic	*GUCA1A*	Het	3	c.149C>T	p.(P50L)	(Downes et al. 2001)
(rs104893968)
**CIC06757**	Ad	*PRPH2*	Het	1	c.514C>T	p.(R172W)	(Wells et al. 1993)
(rs61755792)
**CIC03621**	Ad	*PRPH2*	Het	1	c.1-c581+?del	-	(Boulanger-Scemama et al. 2015)
**CIC00535**	Ad	*PROM1*	Het	10	c.1117C>T	p.(R373C)	(Michaelides et al. 2006)
(rs137853006)

**CIC01196**	sporadic	*PROM1*	Ho	12	c.1354dup	p.(Y452Lfs*13)	(Pras et al. 2009)
**CIC07188**	sporadic	*PROM1*	Het	12	c.1354dup	p.(Y452Lfs*13)	(Pras et al. 2009)
		*PROM1*	Het	IVS 12	c.1454+2T>C	r.(spl?)	(Boulanger-Scemama et al. 2015)
**CIC07045**	sporadic	*PROM1*	Ho	IVS 17	c.1984-1G>T	r.(spl?)	(Boulanger-Scemama et al. 2015)
(rs373680665)
**CIC06642**	Ad	*PROM1*	Het	1	c.7dup	p.(L3Pfs*28)	(Boulanger-Scemama et al. 2015)
**CIC04965**	Ad	*CRX*	Het	4	c.608_609del	p.(S203Ffs*32)	(Boulanger-Scemama et al. 2015)
**CIC03241**	sporadic	*CRX*	Het	4	c.564dup	p.(A189Rfs*47)	Not clear if same mutation as in (Stone 2007)
**CIC3750**	sporadic	*CRX*	Het	3	c.121C>T	p.(R41W)	(Swain et al. 1997)
(rs104894672)
**CIC05218**	Ar	*PDE6C*	Ho	IVS 10	c.1413+3A>T	r.(spl?)	(Boulanger-Scemama et al. 2015)
**CIC02712**	sporadic	*PDE6C*	Het	10	c.1325T>A	p.(M442K)	(Boulanger-Scemama et al. 2015)
		*PDE6C*	Het	10	c.1375C>G	p.(Q459E)	(Boulanger-Scemama et al. 2015)
**CIC06321**	sporadic	*RPGRIP1*	Ho	14	c.2021C>A	p.(P674H)	(Boulanger-Scemama et al. 2015)
**CIC00190**	sporadic	*AIPL1*	Het	5	c.769C>T	p.(L257F)	(Boulanger-Scemama et al. 2015)
		*AIPL1*	Het	5	c.767T>G	p.(I256S)	(Boulanger-Scemama et al. 2015)
**CIC04945**	sporadic	*PROM1*	Het	23	c.2383T>C	p.(W795R)	(Boulanger-Scemama et al. 2015)
		*PROM1*	Het	IVS 13	c.1579-1G>C	r.(spl?)	(Boulanger-Scemama et al. 2015)
**CIC07569**	sporadic	*CRX*	Het	IVS 3	c.252+1G>A	r.(spl?)	(Boulanger-Scemama et al. 2015)
***Other retinal disease genes***
**CIC01571**	Ar	*C2Orf71*	Ho	1	c.2950C>T	p.(R984*)	(Audo et al. 2011) (RP)
**CIC00643**	Ar	*C2Orf71*	Ho	1	c.1949G>A	p.(W650*)	(Boulanger-Scemama et al. 2015)
(rs371289954)
**CIC03112**	Ar	*MERTK*	Ho	17	c.2214del	p.(C738Wfs*32)	(Tschernutter et al. 2006) (RP)
**CIC01242**	Ar	*MERTK*	Ho	3_19	c.483-?_c.3000+?del	-	(Boulanger-Scemama et al. 2015)
**CIC06514**	Ar	*RLBP1*	Ho	7_9	c.526-?_c.954+?del	-	(Boulanger-Scemama et al. 2015)
**CIC03953**	sporadic	*EYS*	Het	11	c.1673G>A	p.(W558*)	(Audo et al. 2010)
(RP)
(rs201823777)
		*EYS*	Het	14	c.2234A>G	p.(N745S)	(Audo et al. 2010)
(RP)
(rs201652272)
**CIC05012**	sporadic	*NMNAT1*	Het	5	c.619C>T	p.(R207W)	(Perrault et al. 2012) (LCA)
(rs142968179)
		*NMNAT1*	Het	5	c.769G>A	p.(E257K)	(Chiang et al. 2012)
(LCA)
(rs150726175)
**CIC06499**	sporadic	*NMNAT1*	Het	5	c.619C>T	p.(R207W)	(Perrault et al. 2012) (LCA)
(rs142968179)
		*NMNAT1*	Het	5	c.769G>A	p.(E257K)	
**CIC05394**	Ar	*RDH12*	Ho	8	c.806_810del	p.(A269Gfs*2)	(Janecke et al. 2004)
(LCA)
(rs386834261)
**CIC07241**	Ar	*RDH12*	Ho	7	c.464C>T	p.(T155I)	(Thompson et al. 2005) (LCA)
(rs121434337)
**CIC07447**	Ar	*RDH12*	Het	8	c.806_810del	p.(A269Gfs*2)	(Janecke et al. 2004)
(LCA)
(rs386834261))
		*RDH12*	Het	8	c.403A>G	p.(K135E)	(Boulanger-Scemama et al. 2015)
**CIC00953**	sporadic	*IQCB1*	Het	6	c.424_425del	p.(F142Pfs*5)	(Otto et al. 2005)
(Senior-Loken/LCA)
		*IQCB1*	Het	8	c.686del	p.(T229Mfs*8)	(Boulanger-Scemama et al. 2015)
**CIC01300**	Ar	*RP1*	Ho	4	c.1719_1723del	p.(S574Cfs*7)	(El Shamieh et al. 2015)
(arRP)
**CIC01380**	Ar	*CRB1*	Ho	11	c.3994T>G	p.(C1332G)	(Boulanger-Scemama et al. 2015) (LCA)
**CIC00963**	Ar	*TULP1*	Ho	11	c.1087G>A	p.(G363R)	(Boulanger-Scemama et al. 2015) (LCA and arRP)
***Lower confidence***
**CIC05007**	Ad	*ROM1*	Het	1	c.339del	p.(L114Sfs*8)	(Boulanger-Scemama et al. 2015) (adRP)

Ar: autosomal recessive; Ad: autosomal dominant; RP: retinitis pigmentosa; MD: macular dystrophy; LCA: Leber congenital amaurosis; [1] personal communication B. Puech. [2] Sergouniotis P. (2012). *Genotype and phenotypic heterogeneity in autosomal recessive retinal disease*. Ph.D. Thesis. Institute of Ophthalmology, University College London, United Kingdom.

**Table 3 ijms-20-04854-t003:** Spectral-domain optical coherence tomography (SD-OCT) abnormalities (N = 49 patients).

	n (%)
***OUTER RETINA***	
EZ irregularities	3 (6)
Hyperreflective layers disruption (foveal/beyond the fovea)	
ELM	39 (80)/27 (55)
EZ	46 (94)/30 (61)
IZ	49 (100)/42 (86)
RPE	0 (0)/0 (0)
Hyporeflective foveal cavitation	3 (6)
Foveal sparing	6 (12)
Hyper-reflective deposits above the RPE	14 (28)
Outer retinal tubulations	1 (2)
Outer nuclear layer atrophy in the macular region	36 (73)
Diffuse outer retinal atrophy beyond the vascular arcades	20 (41)
***INNER RETINA***	
Hyporeflective macular cysts	2 (4)

ELM: external limiting membrane; EZ: ellipsoid zone; IZ: interdigitation zone; RPE: retinal pigment epithelium

**Table 4 ijms-20-04854-t004:** BAF/IRAF abnormalities.

“BAF” Autofluorescence (N = 56 Patients)		“IRAF” Autofluorescence (N = 52 Patients)	
Macular abnormalities		Macular abnormalities	
Minimal alterations, n (%)	6 (10,5)	Minimal alterations, n (%)	6 (11,5)
Loss of foveal hypoAF, n (%)	1 (2)	Loss of foveal hyperAF, n (%)	1 (2)
Foveal hyperAF, n (%)	2 (3,5)	Foveal hypoAF, n (%)	3 (5,5)
Perifoveal hyperAF, n (%)	3 (5)	Perifoveal hypoAF, n (%)	2 (4)
Macular hypoAF, n (%)	11 (19,5)	Macular hypoAF, n (%)	16 (30,5)
Macular hypoAF spots, n (%)	1 (2)	Macular hypoAF spots, n (%)	1 (2)
Diffuse retinal abnormalities		Diffuse retinal abnormalities	
Macular hypoAF + peripheral hypoAF spots, n (%)	20 (36)	Macular hypoAF + peripheral hypoAF spots, n (%)	13 (25)
Macular hypoAF + peripheral “speckled” aspect, n (%)	5 (9)	Macular + peripheral hypoAF, n (%)	16 (31)
Macular hypoAF + peripheral confluent hypoAF patches, n (%)	13 (23)		
without “speckled” aspect, n (%)	6 (11)		
with “speckled” aspect, n (%)	7 (12)		
HyperAF ring		HyperAF ring	
macular, n (%)	9 (16)	macular, n (%)	9 (17)
macular including the optic nerve, n (%)	4 (7)	macular including the optic nerve, n (%)	3 (6)
Peripapillary sparing, n (%), N = 38	6 (16)	Peripapillary sparing, n (%), N = 29	6 (21)

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
