# Peer review of "Phenotype Analysis of Retinal Dystrophies in Light of the Underlying Genetic Defects: Application to Cone and Cone-Rod Dystrophies"

_ijms, 2019, doi:10.3390/ijms20194854_

Round 1
Reviewer 1 Report
The authors addressed most of my concerns. I'm still concerned about sample size, but the authors clearly note that it is a limitation. This information may be helpful to others as validation for their larger (statistically sound) studies. A minor note - "analyzis" should be changed to "analysis".
Author Response
Dear reviewer 1,
Thank you for your comments, with which we fully agree. We changed in the text "analyzis" for "analysis".
We hope that this revised version is now suitable for publication in IMJS.
Yours sincerely,
Elise Boulanger-Scemama
Reviewer 2 Report
I have no additional comments.
Author Response
Dear reviewer 1,
Thank you for your comments, with which we fully agree.
We hope that this revised version is now suitable for publication in IMJS.
Yours sincerely,
Elise Boulanger-Scemama
This manuscript is a resubmission of an earlier submission. The following is a list of the peer review reports and author responses from that submission.
Round 1
Reviewer 1 Report
The manuscript by Boulanger-Scemama, et al., attempts to associate clinical observations of cone and cone-rod dystrophies to genotypic findings. This is an attempt to help select pathogenic variants from genetic screening results of affected patients. The description of the OCT, fundus, and autofluorescence results is thorough and well-described. However, as the authors acknowledge, phenotypic and genotypic heterogeneity is a hallmark of all IRDs. While the study goals are admirable, without statistical support, the results provide little value to the clinical practice. It's understandable that COD/CORD is rare, and patients may willing to participate in such studies may be difficult to find. However, this fact does not warrant publication of an incomplete study. It would be worthwhile to perform power analyses to determine the number of patients need to obtain statistical significance. Additionally, it would be worthwhile to include a control population, such as rod-dominant IRDs like LCA and/or RP. How do your carefully obtained clinical observations compare? Are there findings that may confound one type of disease for another?
Although minor, the abstract appears as if it were written for another journal in another format. It should be re-written for clarity.
Reviewer 2 Report
This article summarizes retinal imaging features along with genetic information in cone and cone-rod dystrophies. This is an important approach to help understanding diseases better and providing precise clinical information to patients when genetic testing is not available. Overall presented data is insightful and the article is well-written. However, most of their graphic presentations are very hard to see.
Specific comments:
1. C2Orf71 is described as C2Orf71 in a text, C2ORF71 in Figure 5. Please check them.
2. Table 2. CIC05853 p.E108K should be indicated p.(E108K) as others, if there is no special reason.
3. Figure 1. Letters in figures are too small to read.
4. Table 3. This table is hard to follow. Please revised.
5. Figure 2. Letters in figures are too small to read.
6. Figure 3. Letters in figures are too small to read.
7. Figure 4. Please consider better presentation. Letters are too small to read.
8. Figure 6. Labels are too small to read.